# Improving Widescale Monitoring of Ectoparasite Presence in Northern Canadian Wildlife with the Aid of Citizen Science

**DOI:** 10.3390/insects13040380

**Published:** 2022-04-12

**Authors:** Emily S. Chenery, Maud Henaff, Kristenn Magnusson, N. Jane Harms, Nicholas E. Mandrak, Péter K. Molnár

**Affiliations:** 1Department of Physical and Environmental Sciences, University of Toronto Scarborough, 1265 Military Trail, Scarborough, ON M1C 1A4, Canada; nicholas.mandrak@utoronto.ca (N.E.M.); peter.molnar@utoronto.ca (P.K.M.); 2Animal Health Unit, Department of Environment, 10 Burns Road, Whitehorse, YT Y1A 4Y9, Canada; maud.henaff@yukon.ca (M.H.); kristenn.magnusson@yukon.ca (K.M.); jane.harms@yukon.ca (N.J.H.); 3Department of Biological Sciences, University of Toronto Scarborough, 1265 Military Trail, Scarborough, ON M1C 1A4, Canada

**Keywords:** parasite, ticks, *Dermacentor albipictus*, winter tick, wildlife health, citizen science, hunting, sampling, surveillance, monitoring

## Abstract

**Simple Summary:**

Surveying ticks on wildlife hosts consistently over time and across space presents many challenges. In Yukon, Canada, the winter tick, *Dermacentor albipictus*, is a blood-feeding parasite that can cause significant losses of hair and blood in moose and other wildlife. The impacts of winter tick infestation in wildlife hosts in this northern region are not well documented. To enhance existing surveillance of winter ticks in Yukon, we implemented a three-year citizen science program, the Yukon Winter Tick Monitoring Project (YWTMP) to engage hunters in the collection of underrepresented moose and caribou samples. Social media, participation incentives, and hide-sampling kits distributed to hunters increased the combined number of annual moose and caribou hide submissions almost 100-fold, and the geographical range of samples by almost 500 km, compared with submission numbers in the previous seven years. Citizen science samples were also used to detect previously unknown infection localities on moose in southeastern Yukon that are spatially separate to known infestations found on elk and deer, helping to build a better picture of infection dynamics on different host animals. Engaging with key demographic groups using structured citizen science programs like the YWTMP can significantly expand sampling efforts in remote areas while maintaining systematic sampling methods to monitor parasites of wildlife health concern.

**Abstract:**

Sampling hides from harvested animals is commonly used for passive monitoring of ectoparasites on wildlife hosts, but often relies heavily on community engagement to obtain spatially and temporally consistent samples. Surveillance of winter ticks (*Dermacentor albipictus*) on moose (*Alces alces*) and caribou (*Rangifer tarandus caribou*) hosts in Yukon, Canada, has relied in part on voluntary submission of hides by hunters since 2011, but few samples were submitted. To enhance sampling efforts on underrepresented moose and caribou hosts, we implemented a three-year citizen science program, the Yukon Winter Tick Monitoring Project (YWTMP), to better engage with hunters in hide sample collection. A combination of in-person and social media outreach, incentivized engagement, and standardized hide sampling kits increased voluntary submissions of moose and caribou hides almost 100-fold since surveillance began. Citizen science samples expanded the northernmost geographic extent of existing sampling efforts for moose by 480 km and for caribou by 650 km to reach 67.5° N latitude. Samples also resulted in new detections of winter ticks on moose hides that are spatially separate to those submitted for other cervids in Yukon. Findings from the YWTMP have provided an essential baseline to monitor future winter tick host–parasite dynamics in the region and highlighted priority areas for ongoing tick surveillance.

## 1. Introduction

Engaging non-scientists in tick surveillance has demonstrated benefits to public health in obtaining new records of tick detection [1,2], assessing the risks of tick-borne illness [3,4,5], and educating communities on tick-bite prevention [6]. Targeted engagement of key community demographics at higher risk of contact with ticks, such as school children [7,8], hikers, hunters, and outdoor enthusiasts [9], and those with a general interest in tick monitoring in their community [3,6,10], not only provides high-quality data for modelling and mapping tick distributions and that of their associated pathogens, but may also build long-term relationships of benefit to researchers and citizen participants alike [11,12,13]. Many examples of citizen science-led tick monitoring to date have focused on passive collection of samples encountered by participants, such as mail-in tick schemes aimed at describing tick-borne disease risk to humans in the U.S.A. [3,10,14], Canada [4,6], and Spain [9], and reporting tick presence on domestic horses in Canada [15]. In most cases, ticks collected in these programs are a direct result of the tick’s natural host-seeking process, with submissions primarily from humans or domestic animals or from peri-urban environments. Monitoring ticks that primarily infect wildlife species, as opposed to humans, presents additional challenges to both sampling methodology and community engagement, owing to inherent logistical barriers for collection.

The nature of on-host sampling means that tick detection is necessarily limited by access to physical host specimens, which may be particularly challenging to obtain when assessing presence in vast and remote areas. Surveillance of ticks on large game species, such as cervids, has a reasonably standardized methodology, commonly using set time- or length-based searches across transects in the hair of living or dead host animals [16]. These methods are increasing in popularity for ectoparasite monitoring, being less time consuming and labor intensive than hide digestion techniques, which destroy the hide and require laboratory processing using chemical agents [17,18]. Additionally, conducting visual searches of hides for ticks can be carried out in situ at hunter check-stations or on live-captured hosts, or from samples of hides taken from hunted or roadkill animal carcasses [17,19,20,21,22].

In northern Canada, where communities continue to engage in hunting for both food and traditional cultural purposes, community-based integration of hunters in wildlife health surveillance has demonstrated its value for both researchers and community members alike [11,23]. Submission of the hides of harvested moose (*Alces alces*) and caribou (*Rangifer tarandus*) from the Sahtu Settlement Area in the Northwest Territories confirmed the presence of the winter tick, *Dermacentor albipictus* [24,25], a wildlife parasite of particular concern for cervid health, in this northern region. Winter ticks are blood-feeding ectoparasites that have been implicated in mass die-offs of moose in North America [26,27,28,29], including ongoing declines in local moose populations in the U.S. [30]. Burdens upwards of 50,000–60,000 ticks per moose may result in host death due to several combined factors, including hair loss, blood loss, and reduced foraging behaviors [29,30,31,32]. Winter tick impact on other cervid species has historically been considered less severe [33,34], although it is also less studied. Recent reports raise concerns over the potential impact of winter ticks on caribou [35,36], elk (*Cervus elaphus canadensis*) [37,38], and white-tailed deer (*Odocoileus virginianus*) [39]. Surveillance of winter ticks on all cervid species is important for understanding their distribution among host populations and in monitoring their effects. Although widely used in southern Canada and the U.S., the use of hunter check stations in remote northern Canada is not a practical or reliable method for collecting samples. Further, due to the life cycle of the tick, checking hides in the field in the early part of the hunting season (i.e., September and October) is challenging as the ticks are still in their larval form (<1 mm long). This means that submission of hide samples by hunters for thorough visual checks using laboratory tools likely improves detection, and therefore provides a consistent and long-term means of monitoring winter ticks in the North [17,22].

The Yukon Winter Tick Monitoring Project (YWTMP) was a collaboration between the Yukon Government’s Animal Health Unit (AHU) and researchers at the University of Toronto. Established in 2018, the project sought to enhance existing territorial monitoring of winter ticks on cervids by expanding AHU’s hide submission scheme and to complement field-based sampling efforts for larval winter ticks off-host [40]. Government monitoring of winter ticks on hides in the territory began in 2011, following detection of the parasite on a managed population of elk [41]. Hides from roadkill, illegal kills, conflict kills, and animals found dead are also submitted voluntarily as part of this scheme but, unlike the annual harvest, are necessarily opportunistic in nature. All hides are sampled for winter ticks according to a standardized “hair transect” protocol commonly used for checking ectoparasites on cervids [17,42] (and see Materials and Methods, Section 2.4 Sample processing). To explore if winter ticks in Yukon are solely maintained by elk and to examine the potential spread of the ticks to other host species, the hide sampling program also includes harvested moose, caribou, and deer, to be submitted on a voluntary basis.

Voluntary participation in scientific research has generally been split into two main categories of motivation: intrinsic (interest or enjoyment driven), and extrinsic (outcome or reward driven) [43,44]. A call for voluntary submissions of moose and caribou hides was included in the Yukon Hunting Regulations Summary starting in 2014, but few samples were received each season (1 April–31 March), despite the majority of hunters commonly holding a seal for one or both of these species [45]. The low number of voluntary moose and caribou hide submissions to the AHU scheme from 2011 to 2017 indicated that engagement from the hunting community was low if only intrinsic factors were considered. This was likely due to the considerable effort of bringing the large, heavy hide from the field, particularly in cases where the hunter would not normally keep this part of the animal.

We sought to determine whether changes to the existing hide-submission program could increase voluntary moose and caribou hunter participation across Yukon, as a means of improving detections of winter ticks in the territory. For this study, we do not include data on voluntarily submitted animals that were killed by non-hunted means (e.g., roadkill, found dead), but focus solely on engagement with the hunting community. The YWTMP study took place over three hunting seasons (2018–2020) and was designed to appeal to volunteer motivations, offering incentives and simplified hide sample submission kits that reduced participant burden. Success of the scheme was evaluated based on the number of hides or samples returned by hunters across each hunting season in the study (2018–2020) relative to previous engagement in prior seasons (2011–2017). We also considered the level of conformity with hide sample kit submissions as an indicator of success. To determine the efficacy of this approach for widescale monitoring across the territory, we compared the total number and geographic location (Game Management Subzone, GMS) of samples received before and after YWTMP scheme implementation and their winter tick infection status. In presenting these findings, we show how increased engagement with the hunting community to boost sample numbers can supplement existing monitoring efforts and, critically, improve knowledge of tick distribution on hosts at a regional level.

## 2. Materials and Methods

### 2.1. Voluntary and Mandated Hunted Hide Submission

The Yukon government’s hide submission program began in 2011 and continued throughout the duration of the YWTMP study, processing mandatory harvested hide submissions for elk and, from 2018 onwards, mule deer (*Odocoileus hemionus*). Moose and caribou samples, on the other hand, continue to be submitted only on a voluntary basis. Hides of mandatory submissions are required to be taken to a Department of Environment office within 15 days of the harvest [46]. All hides can be returned to the owner post-sampling if requested.

### 2.2. Hide Incentives Program

Beginning in August 2018, hunters were offered their choice of an incentive (a stainless-steel thermal flask or two high-quality game meat bags) for every hide sample submitted to the YWTMP scheme during that season. The scheme was advertised through the Yukon Hunting Regulations, during the government’s Hunter Education and Ethics Development (HEED) course, via YWTMP social media posts (Facebook, Twitter), and through printed material in Department of Environment and First Nations offices and local businesses throughout the territory (Figure 1a). Incentives were received by participants at the time of sample submission to a Department of Environment office. The incentives scheme also applied to the 2019 and 2020 seasons for participants returning full moose and caribou hides or samples for these species, as part of YWTMP hide sample kits.

### 2.3. Hide Sample Collection Kits

To increase ease of collection for participants and encourage consistency in sample collection, we designed a relatively small, lightweight sampling kit that could be taken into the field by the hunter (Appendix A). Winter ticks are found at the highest densities in the neck and shoulder region of their hosts [34,47], and this body region has been shown to provide a suitable location for tick detection on moose [17,34]. We therefore followed methods of collection and visual inspection adapted from Sine and colleagues’ [17] standardized collection methods, requesting one 20 cm × 40 cm hide sample from the right shoulder of the animal. The size of the hide sample was chosen because it balanced ease and therefore portability of collection with detection probability. The sample was collected from a standardized location where winter tick abundance is generally highest [34]. Sample kits consisted of a large Ziploc™ (SC Johnson, Bay City, MI, USA) plastic bag (26.8 cm × 27.3 cm) that contained a single piece of paper printed with written and visual instructions that also doubled as a hide sample template (20 cm × 20 cm) (Figure 1b), and a pair of single-use nitrile gloves (primed Prima Touch^®^ Nitrile Extra Strong). Each kit was labelled with a unique identifier and had fillable form fields designating hunted species (moose or caribou), sex, and kill date and locality. Participants were asked to cut one sample of hide, approximately twice the size of the template, from their kill and to return it inside the Ziploc^TM^ bag to any Department of Environment office. No personal or identifying information relating to the participant was collected. Hunters were requested to collect the hide sample within six hours of a kill to reduce the likelihood of ticks leaving the dead host [17] and advised that samples that could not be submitted within 24 h should be frozen. Care was taken to remind participants that larval winter ticks may not be easily visible to the naked eye and to submit samples even if they appear to be tick-free. Although winter ticks are infrequently known to bite humans [32,44], hunters were verbally reminded during kit pick up to conduct a tick-check on themselves and to wear the gloves provided when handling the hide.

Physical kits were distributed via Department of Environment offices in Whitehorse, Haines Junction, Watson Lake, Teslin, Mayo and Dawson, the Yukon Fish and Game Association office in Whitehorse, and several First Nations harvest offices throughout the southern region of Yukon. Double-sided informational postcards accompanied the kits showing all winter tick life stages and additional information on the species. Additionally, hide-sample templates and winter tick information cards were available to download in e-copy via the YWTMP Facebook page (https://www.facebook.com/tickymoose). Kits were available throughout the moose and caribou hunting seasons of 2019 and 2020 (September to December) and samples were encouraged to be returned within this same timeframe. Although accepted throughout the year, no kits were returned past December each year.

### 2.4. Sample Processing

Comparable protocols were used to sample both full hides and YWTMP hide samples for winter ticks (Appendix A). Full hides (both mandatory and voluntary submissions) were assessed using a standard line transect method for surveying ectoparasites on hides, following a protocol based on Sine and colleagues [17,42]. Hides were laid on a flat surface and five equal transects 70 cm long and spaced approximately 2.5 cm apart were taken on either side of the midline, running from the neck, down the shoulders and back. Transects were measured using a flat meter rule and the hair parted to the skin using a knitting needle, along which ticks were removed and counted. The total number of ticks were recorded by life stage along each transect and summed to record the total number of ticks per hide. In cases where a full hide could not be processed immediately, it was stored frozen at −20 °C and left to thaw for up to 24 h before sampling. Hide samples from YWTMP kits were immediately frozen on receipt and left to thaw at room temperature for approximately eight hours prior to processing. The length and width of each hide sample was recorded, before being separated into transects approximately one cm apart and running the full length of the sample. Hide samples were first placed under a magnification lamp (Intertek GS-T00589) and, as with the full hide transect method, a knitting needle was inserted under the hair along each transect to record and remove ticks. All sample bags were also checked for loose ticks before disposal, but none were found. In all cases, a subset of ticks was identified via microscopy (Olympus SZ61) to species based on morphological characteristics as given in Lindquist et al. [48], and specimens archived within the Animal Health Unit’s collection.

### 2.5. Evaluating YWTMP Success

Defining what constitutes success in citizen science projects is challenging, but commonly used indices are participant numbers and ongoing commitment to the project over time [49,50]. In addition to the number of hunters engaging (taking hide kits) and actively participating (returning samples), we also considered the level of conformity with hide sample kit instructions as an indicator of successful participant engagement. Submissions were scored based on two components of the kit: (1) size of hide sample submitted; and (2) completeness of information as requested on the sample label. For each kit returned, hide samples were measured (length and width, cm) and sizes were converted to area measurements (cm^2^). These measurements were scored according to their closeness to the requested sample size as per the template provided (800 cm^2^), with samples between 700 and 900 cm^2^ receiving a score of 1, and all others scored as 0. Label information included kill date, species, and sex (each scoring 1 if complete, 0 if blank), and kill location. Localities recorded as point locations as requested were scored as 2; those with descriptive locations that allowed us to later estimate their coordinates were scored as 1; all others received 0. We used the final sum of scores across all categories to assess compliance level for all participants, with a maximum achievable score of 6 indicating a near-perfect hide sample size and all information exactly as requested. All statistical analyses were carried out using R statistical software (version 4.1.1, R Core Team, 2021), and locations mapped according to GMS using GIS (QGIS v.3.16.10, QGIS Core Development Team, 2020).

## 3. Results

No species of tick other than *D. albipictus* were found among hide samples from any of the submission methods.

### 3.1. YWTMP Engagement and Participation

One full moose and one full caribou hide were received for sampling during the 2018 season (Figure 2), accounting for approximately 0.1% of moose and 0.4% of caribou harvested by licensed hunters over this time (Table 1 and Figure 3). Approximately 8.5% of all licensed hunters took a hide sample submission kit for the 2019 season (*n* = 435/5135), of which approximately 10% were returned with samples (*n* = 44 kits) (Table 2). A total of 30 full hides and 48 hide samples (4 partial, no kit) were received over the 2019 season, comprising 53 moose and 25 caribou (Figure 2). Submissions accounted for approximately 7.5% of the total moose and 5.5% of the total caribou harvested during the 2019 season, up from an average from 2011 to 2017 of 0.1% for both moose and caribou respectively (Table 1 and Figure 3). The 2020 season saw 18 full hides and 58 hide samples submitted, of which there were 51 moose and 25 caribou. These submissions accounted for approximately 9% of moose and 8.5% of caribou reported to have been killed by licensed hunters during this season (Table 1 and Figure 3).

### 3.2. YWTMP Hide Sample Kit Conformity

Variability in completeness scores was similar within hunted species and between years, though generally higher among moose hunters than caribou hunters (caribou: coefficient of variation (CV) 2019 15%, 2020 14.9%; moose: CV 2019 22.2%, 2020 24.9%). In both 2019 and 2020 sampling years, all sample submissions contained complete information on kill date, host species, and sex.

In 2019, the average area of returned hide samples in the sample kits was 805.2 cm^2^ (SD ± 250.7 cm), with a mean width of 25 cm (SD ± 4.6 cm) and length of 32.4 cm (SD ± 8.4 cm) compared with the 800 cm^2^, 20 cm × 40 cm size requested. Twenty-two participants submitted kill locality coordinates, 20 gave detailed locality information that allowed for estimation of a point location, and two participants returned inadequate locality information or none. Five out of 44 (13.6%) submissions fully met exact submission specifications (hide size, location coordinates), receiving a perfect score of six, compared with a mean completeness score over all submissions of 4.5 (range 3:6, SD ± 0.7) (Figure 4a).

In 2020, the average area of hide samples was 669.2 cm^2^ (SD ± 204.1 cm), with a mean width of 23.7 cm (SD ± 4.7 cm) and length of 28.6 cm (SD ± 8.4 cm). One sample was excluded as it was clearly not from the body region (shoulder) requested. Of the remaining 55 samples, locality information in the form of coordinates was submitted by 23 participants, with 19 providing detailed information that allowed for estimation of a point location. Thirteen participants gave locality information that was too vague or missing. On average, samples in 2020 received a score of 4.5 out of 6 (range: 3:6, SD ± 0.9) with 7 of the 56 (12.5%) submissions fully meeting the requested specifications (Figure 4b).

### 3.3. Distribution of Winter Ticks on Hunted Cervids in Yukon

Winter ticks have been recovered from hunted moose, elk, and mule deer hides in Yukon since sampling began in 2011 (Figure 5). Prior to the YWTMP scheme, winter ticks had been found on one or more hunted hides submitted from seven of nine sampled GMSs, ranging from latitudes 60.1–61.7° N and longitudes 128.6–137.8° W. Tick detections on hunted animals were mostly on mandatory submissions of elk and mule deer (*n* = 28 elk, *n* = 12 deer), with few detections on moose (*n* = 2). No ticks were found on hunted caribou prior to the YWTMP scheme. During the first season of the YWTMP in 2018, winter ticks on hunted animals were only found on elk (*n* = 4) and in only three of six GMSs. During the 2019 hunting season, YWTMP samples of moose and caribou were received from 46 GMSs, including 42 subzones that had not previously been sampled, and ranging from latitudes 60.0–67.5° N and longitudes 128.5–139.9° W, an increase in the northern limit sampled since 2011 of 480 km for moose and 650 km for caribou. In 2020, the overall geographic range of YWTMP samples was similar to the previous year with latitudes ranging from 60.0–67.1° N and longitudes 128.5–139.7° W within 49 GMSs. Of these, 29 subzones were included that had not been sampled in any previous year. This now brings the cumulative number of GMSs in the territory that have received hide submissions (including for elk and deer) to 86, representing an almost ninefold increase in sampling locations achieved in two years. Winter ticks were found on hunted moose (*n* = 3) in three of these newly sampled subzones (Figure 5a), in addition to continued detections from mandatory submissions of hunted elk (*n* = 8) and mule deer (*n* = 5) (Figure 5b). These new detections indicate winter tick presence on moose that is spatially separated by over 450 km from the managed elk core range.

## 4. Discussion

We found that widescale monitoring of winter ticks at the territorial level in Yukon was significantly improved through voluntary citizen science participation by hunters. The highest level of engagement in sample submission occurred when hunters were provided with simplified sampling kits and incentivized returns, compared with voluntary submissions alone. These findings are broadly in agreement with other studies examining participant motivation, where ease of contribution and recognition of its value to ongoing research are considered important factors driving engagement numbers and volunteer retention [43,44].

Due to differing aims and objectives across citizen science projects, there are no standardized measures used to define success [49,50]. Our project first and foremost aimed to increase the number and geographic range of sample submissions for monitoring and surveillance of winter ticks on potential hosts, and so, in this regard, the program may be considered successful given the sharp increase in hide submissions over its duration. However, the overall spatial range covered by these samples is still inconsistent across large parts of Yukon, and hide numbers represent only a small proportion of the total number of moose and caribou hunted in the territory each season. Although we saw a significant increase in engagement compared with the previous eight years of the voluntary scheme, the total number of submissions suggests that most licensed hunters (~90%, Table 2) did not participate. We did not collect data on hunter demographics during this study that might indicate whether, for example, non-resident hunters may be less likely to engage with sample collection than residents. Similarly, due to anonymity of submissions, we cannot evaluate whether the same hunters are more likely to return hides or samples on an annual basis. It seems likely that, as in other citizen science and volunteer engagement schemes, the majority of participation will be by a passionate few [44], represented here by the approximately 10% of hunters intending to hunt moose or caribou (with seals) that participated in the program (Table 2).

The economic cost of running incentivized public engagement programs may be a potential limitation in the short term, though they frequently generate data or knowledge that would otherwise take years to accumulate [12]. Although commonly used in marketing surveys, providing incentives to participate in science-based research may still be considered controversial, though concerns are generally levelled at studies for which human participants are themselves the source of data collection, such as in medicine or social sciences [53]. Studies on participatory engagement have noted that receipt of some form of recognition for their effort is not an uncommon expectation among volunteers [43,54] and may serve to improve rates of response [55]. Based on the limited participation in the first year of the YWTMP in 2018, we observed that incentives alone did not appear to be the sole motivating factor in hunter engagement. Reduced burden, in the form of simplified sample collection, may have played an important role in determining participation. Although the convenience of submitting a smaller hide sample appeared popular, accounting for over half of all submissions in 2019 (*n* = 44/79), more hunters chose to submit full hides for winter tick analysis during that season than in any previous year. Full hides submitted by moose hunters accounted for 20% of all moose submissions (*n* = 11/53), three times the number received voluntarily from 2011 to 2017 inclusive (Figure 3). From caribou hunters, this number was even higher, with 19 full hides submitted for winter tick checks in 2019 (76% of all samples, *n* = 19/25; Figure 3), and 15 full hides in 2020 (60% of all samples, *n* = 15/25). It seems likely that both the 2019 and 2020 seasons benefitted from an increased awareness of the YWTMP incentive scheme among hunters since it was first advertised in September 2018 and that time-to-engagement may have been delayed. However, it is not possible to fully disentangle the effect on engagement based on reward versus concerns for cervid health. Increased knowledge of winter ticks and their potential harm to hosts, gained from YWTMP social media and printed materials, may have served to improve participation in the scheme over time. Recognizing the value of community contributions, particularly in cases where participation requires considerable effort in the collection, transportation, and submission of physical samples, not only serves to partly compensate volunteers for their time and efforts but emphasizes the importance of the data collected as part of a larger research program and improved winter tick surveillance. These findings suggest that appealing to the extrinsic motivations of voluntary participants may be particularly worthwhile to boost sample receipt over time, but we suggest that this may be most effective in conjunction with improved outreach that equally appeals to intrinsic concerns. In our case, alternative options to immediate physical rewards, such as prize draw entries, could potentially optimize financial costs in future while building capacity for engagement through improved outreach programs in remote areas [54,55].

The impact on data quality is one concern that has been raised with regards to engagement of non-scientists in research projects [56,57]. We found that the overall level of conformity with sampling method and data collection requests, although not perfect, was still high. Size of hide samples had the greatest variability and was likely due to challenges in accurately cutting hides in the field, and only three out of 44 samples were too small to trust transect findings. Obtaining kill locality data is often difficult among hunters who wish to protect the knowledge of prime hunting locations, even for the purposes of scientific research. By way of comparison, a similar incentivized, hunter-based sample scheme run in the neighboring Northwest Territories by Kashivakura and colleagues from 2010 to 2012 also found relatively high levels of missing locality data in their submissions [24]. Repeated assertions of data confidentiality and personal anonymity throughout the YWTMP, and association with a known, long-term partner (territorial government), may have assisted in building participant trust in this regard and resulted in the relatively high number of point locations (latitude and longitude, and GPS coordinates) provided. Despite uncertainties in precision, these data are still valuable for ongoing research, as they can be used to assist in determining more focused field sampling locations for off-host studies of larval ticks, for which small spatial scales can be important [58,59].

Data from hunted sources may include biases with respect to sample demographics (age, sex), collection time (hunting season) and geographic locations selected (permit hunt areas; hunting exclusion zones) [60,61]. Hunting season for moose and caribou in Yukon coincides with the breeding season for both moose and elk and is also the same period that winter tick larvae are actively seeking a host [39]. The large majority of hunted cervids are adult males, with very few female animals hunted and only by First Nations communities as part of traditional practice [52]. In Maine, U.S.A., a survey of winter ticks on moose hunted in October found that male moose may contain both greater abundances and ticks at a higher stage of development than female moose or calves [62]. This suggests that, although our detections of winter tick presence are unlikely to be negatively affected by biases in selection by sex, there may be limits in the inferences regarding winter tick prevalence at the population level.

Informal discussions throughout the YWTMP scheme with local hunting groups, associations and First Nations residents indicated that, to date, hunters have rarely interacted with ticks in Yukon. One of the benefits of the YWTMP has therefore resulted from outreach with community members, highlighting tick presence in Yukon that was not previously common knowledge for some hunters. Although winter ticks do not frequently bite people, they are generalists and will attach to a wide range of hosts [48]. Wildlife health remains of primary concern with respect to this species, but a growing body of evidence suggests that, in some cases, winter ticks may also be able to vector diseases communicable to humans [63,64]. Additionally, increasing public awareness of ticks of all species in northern communities, particularly among groups at high risk of contact such as hunters, may serve to improve both tick reporting outside of the YWTMP and commitment to tick bite prevention practices in future [6,8,65].

Overall, the absence of winter ticks detected on samples from above 62° N in Yukon remains consistent with past records, with all anecdotal reports of moose with distinctive, likely winter tick-induced hair loss from below this latitude [66]. However, the inclusion of voluntarily submitted hides of hunted moose and caribou changes our perception of the current spatial distribution of winter ticks on Yukon hosts when compared with mandatory submissions alone. The increased number of hide submissions, while providing a relatively low sample size overall, has revealed three new localities of winter ticks on moose in the Liard region of Yukon, around Watson Lake, thus, beginning to build a clearer picture of winter tick activity in this southeastern region of the territory. Without these additional samples, the only other confirmed detections of winter ticks on hunted hosts and from field studies are in and around the Ibex Valley, approximately 40 km northwest of Whitehorse, in managed elk core habitat (Figure 5b). The large distance between these areas of detection (~450 km) indicates that it is unlikely that moose have become infected with winter ticks from elk and may represent range expansion of ticks from British Columbia where they are found on moose [67] and caribou [36]. Understanding the origin and likely host interactions with ticks in these regions is critical for pinpointing locations of interest for future monitoring of cervid health and provides baseline information from which to assess changes to winter tick distribution in future [25,68].

## Figures and Tables

**Figure 1 insects-13-00380-f001:**
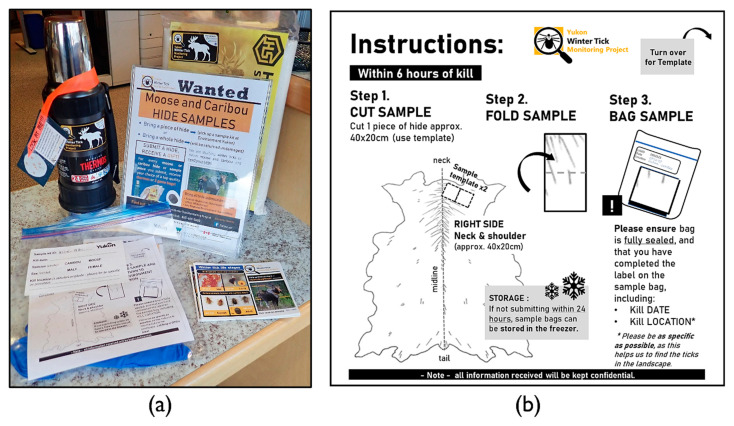
Yukon Winter Tick Monitoring Project (YWTMP) hide submission scheme materials. (**a**) Display at Department of Environment license and permit desk, Whitehorse. Incentives were displayed along with sample kits for collection and informational materials on winter ticks. (**b**) Front side of hide sample template with instructions for collection and storage, as included in each sample kit (see also Appendix A).

**Figure 2 insects-13-00380-f002:**
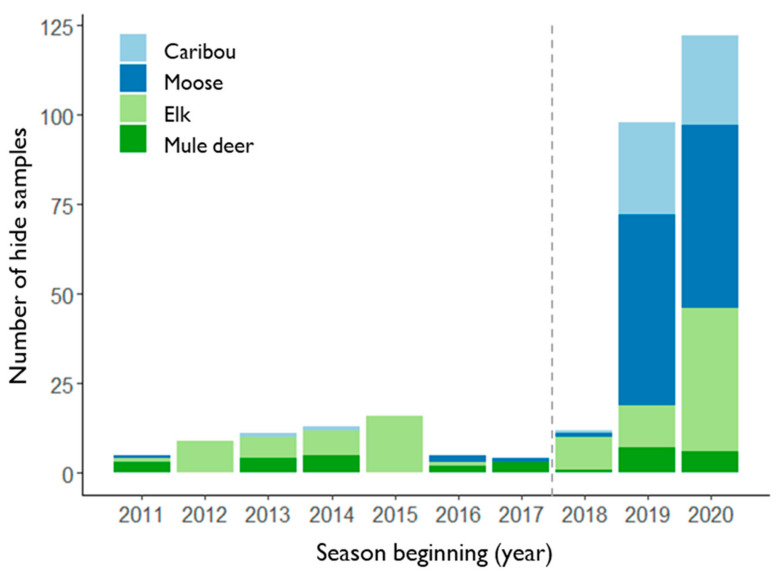
Total number of hunted hide and hide sample submissions per annual hunting season (1 April–31 March), grouped by species (shaded, stacked bars). The Yukon Winter Tick Monitoring Project scheme first came into effect within the 2018 season, indicated by the grey dotted line. Hide sample submission kits were available from 2019 onwards.

**Figure 3 insects-13-00380-f003:**
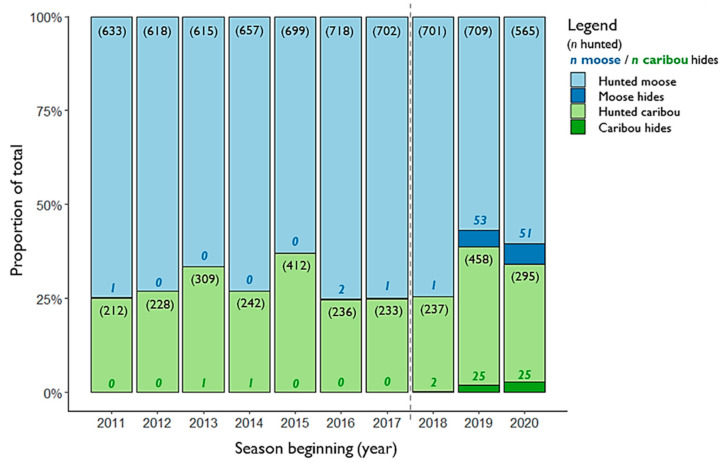
Proportion of the number of moose and caribou reported harvested relative to the number of hide samples voluntarily submitted for each species each season, 2011–2020. Numbers in parentheses refer to the number of animals reported in the licensed big game harvest each year; bold numbers are the total number of moose (shaded blue), and caribou (shaded green) hides received that season. Years that the YWTMP was active are indicated to the right of the grey dotted line.

**Figure 4 insects-13-00380-f004:**
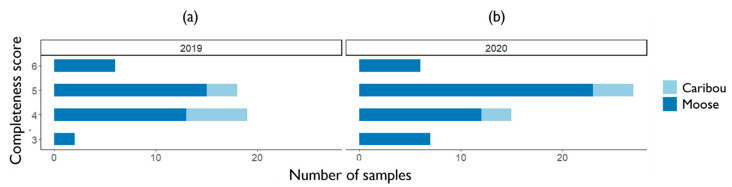
Level of hide sample kit conformity for moose and caribou in (**a**) 2019 and (**b**) 2020, as measured by completeness scores (see Section 2.5 for description of scoring method).

**Figure 5 insects-13-00380-f005:**
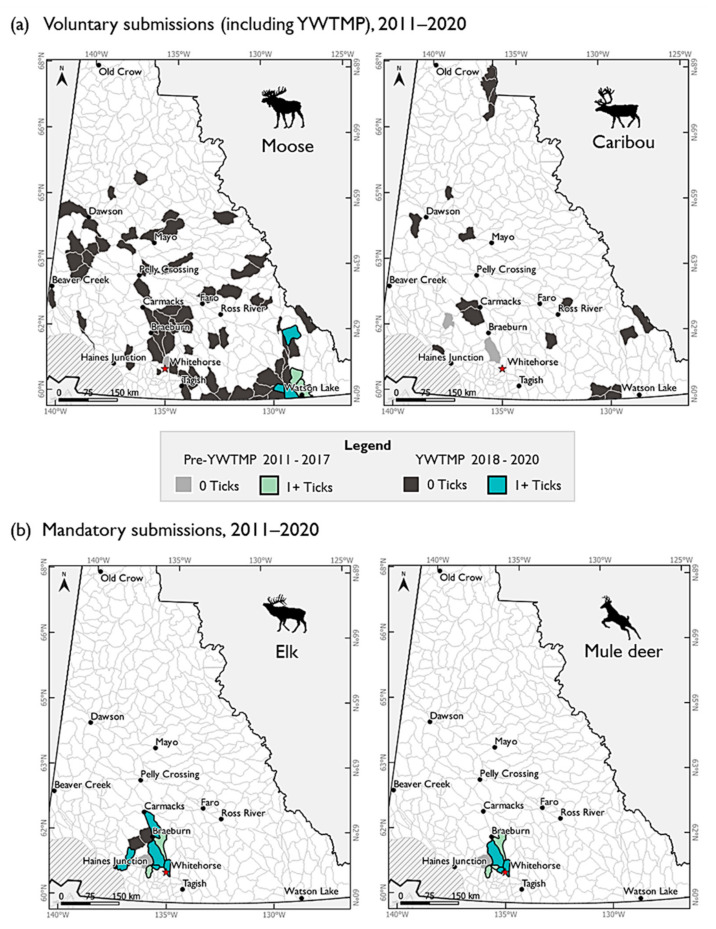
Game Management Subzones (GMS) in Yukon where hunted cervid hides have been received through either (**a**) voluntary, or (**b**) mandatory sources, as dictated by hunted species: moose, caribou, elk, and mule deer. Shading indicates where one or more hide samples have been received per subzone, and their status: winter ticks present, or absent. In cases of multiple samples per subzone over time, shading relates to positive detection history (e.g., a location with a tick-positive sample in 2012, but a subsequent negative detection in 2018, will still be shaded as a tick-positive GMS). White subzone areas have not been sampled. Note: hides from roadkill, illegally killed, conflict kill, and animals found dead are not included here. For a full map of all species, see Appendix A.

**Table 1 insects-13-00380-t001:** Number of moose and caribou hides received (2011–2020), as a percentage (%) of the total number of animals of each species that were reported as part of the licensed game hunt each season. Values for seasons during which the Yukon Winter Tick Monitoring project was active (2018–2020, shaded) are given in bold.

Harvested Species	2011	2012	2013	2014	2015	2016	2017	2018	2019	2020
Moose	0.2	0	0	0	0	0.3	0.1	**0.3**	**7.5**	**9.0**
Caribou	0	0	0.3	0.4	0	0	0	**0.8**	**5.5**	**8.5**

**Table 2 insects-13-00380-t002:** Number of hide sample kits distributed and returned relative to licensed hunting statistics for Yukon each season. Note: hide sample kits were not available in 2018. *Engagement: All hunters* = kits distributed/hunting licenses, *With seals* = kits distributed/mean (moose + caribou seals), *Successful harvest* = kits distributed/harvested animals; *Participation* = kits returned/kits distributed. Licensed hunters must have a seal for the species they intend to harvest that season. Hunters can hold seals for different species; not every hunter with a seal will make a kill. Harvested animal numbers as reported in the annual Yukon Hunting Regulations Summary for 2020–2021 [51] and 2021–2022 [52]; number of hunting licenses and seals issued each year provided by Yukon Department of Environment (unpublished data).

Number of:	2019	2020
Hunting licenses	5135	5125
Hunting seals issued—Moose	3957	3950
Hunting seals issued—Caribou	3467	3668
Harvested animals (Moose + Caribou)	1167	860
Kits distributed	435	617
Kits returned	44	56
**Engagement**		
All hunters	8.5%	12.0%
With seals (average)	11.7%	16.2%
Successful harvest	37.3%	71.7%
**Participation**	10.1%	9.1%

## Data Availability

Data available in a publicly accessible repository. The data presented in this study are openly available in Figshare at https://doi.org/10.6084/m9.figshare.19249064.

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
