# Peer review of "Improving Widescale Monitoring of Ectoparasite Presence in Northern Canadian Wildlife with the Aid of Citizen Science"

_insects, 2022, doi:10.3390/insects13040380_

Round 1

Reviewer 1 Report

Well written manuscript. Clear presentation of the research goal and methodology. Figures are easy to interpret and map format is useful. Very small amount of copy editing required-minor.

Author Response

We thank Reviewer 1 for their positive review of our manuscript. We have reviewed the paper for readability and corrections as suggested, and made a few minor improvements to the text (please see tracked-changes).

Reviewer 2 Report

This paper describes a novel, citizen science approach to use/transform hunters from passive to active data collectors, with the goal of improving detection of an important ectoparasite (winter tick) of multiple wildlife/big game species.  In the context of climate change, the importance of this work is magnified given the increasingly frequent and deleterious impacts of winter tick across the NA range of moose.  The basic justification/objective is to address lack of information and improve data collection in remote areas by engaging resource users. The paper is well organized and written.

A few specific comments:

1) Line 74: I would include Sine et al. (Ref. #17) in this list (19-21) as well as Bergeron and Pekins (2014 - Alces 50: 1-15) that is not listed as a reference.  Combined, these two studies provided the foundation and verification of measuring winter ticks on harvested moose and establishing a useful index. The first was the initial study and the second verified and extrapolated its use as an index for management which is a logical next step for using hide samples and measurements. Perhaps both papers should be cited on Line 98 because both concluded that point (long-term monitoring), specifically the latter.  See next comment -  

2) Line 95: I don't think it accurate to describe larvae as "nearly undetectable". Your lab measurement/technique is the same as done on harvested moose at hunter check stations that mostly involve larval counts.  These are the same hide measurements done by Sine and Bergeron, so you are indirectly/inadvertently questioning your own approach/accuracy unless hide samples were dominated by larger nymphal ticks. Albeit, this is more important relative to developing a numerical index versus identifying presence, but "undetectable" seems overly conservative and a poor word choice.  Maybe "challenging"? Isn't the reason for requesting hide samples logistical (biologist unavailable) not an inability to verify/identify the presence of larvae on a hide?

3) The question is why don’t more hunters collect the sample?  Although it is recognized that higher incentive might increase participation (Lines 397-399), I wonder if a better incentive isn't elevating the concern about the resource versus a prize?  After all, the animal likely represents the largest value to the user, so perhaps the potential harm to that resource from winter ticks is not understood?  Perhaps a suggestion to focus education (marketing) in combination with "better" reward, and targeting hunters in "unsampled" areas?  See next comment -

4) Lines 433-439: Is there really a legitimate concern about human health and winter ticks and is this a reasonable/honest incentive to promote participation versus 3) above?  This seems a stretch given the remoteness of the area, the probability of human exposure, and that exposure is strictly limited to the questing period of 8-10 weeks in autumn only. 

Author Response

We thank Reviewer 2 for their positive review of our manuscript and helpful comments, which we respond to below -

A few specific comments:

1) Line 74: I would include Sine et al. (Ref. #17) in this list (19-21) as well as Bergeron and Pekins (2014 - Alces 50: 1-15) that is not listed as a reference.  Combined, these two studies provided the foundation and verification of measuring winter ticks on harvested moose and establishing a useful index. The first was the initial study and the second verified and extrapolated its use as an index for management which is a logical next step for using hide samples and measurements. Perhaps both papers should be cited on Line 98 because both concluded that point (long-term monitoring), specifically the latter.  See next comment -  

Thank you for this suggestion.  We agree it makes sense to include these references for the reasons you mention – now included on lines 74 and 98 of the manuscript, and bibliography updated to include Bergeron & Pekins 2014.

2) Line 95: I don't think it accurate to describe larvae as "nearly undetectable". Your lab measurement/technique is the same as done on harvested moose at hunter check stations that mostly involve larval counts.  These are the same hide measurements done by Sine and Bergeron, so you are indirectly/inadvertently questioning your own approach/accuracy unless hide samples were dominated by larger nymphal ticks. Albeit, this is more important relative to developing a numerical index versus identifying presence, but "undetectable" seems overly conservative and a poor word choice.  Maybe "challenging"? Isn't the reason for requesting hide samples logistical (biologist unavailable) not an inability to verify/identify the presence of larvae on a hide?

We are happy to change our phrasing here, we were just trying to convey the difficulty of detecting larvae due to their small size. We have now updated the relevant sentence (lines 93-95) as follows:

“Further, due to the life cycle of the tick, checking hides in the field in the early part of the hunting season (ie. September and October) is challenging as the ticks are still in their larval form (<1mm long).”  

3) The question is why don’t more hunters collect the sample?  Although it is recognized that higher incentive might increase participation (Lines 397-399), I wonder if a better incentive isn't elevating the concern about the resource versus a prize?  After all, the animal likely represents the largest value to the user, so perhaps the potential harm to that resource from winter ticks is not understood?  Perhaps a suggestion to focus education (marketing) in combination with "better" reward, and targeting hunters in "unsampled" areas?  See next comment –

This is an interesting question, and one that probably has more than one answer. In this paragraph we are talking specifically about extrinsic motivational factors, as opposed to intrinsic concerns. But we agree with your point that the motivation to return a sample could likely be improved through an increased understanding of potential impacts on moose and other host species.

We have brought this point into the discussion, from lines 391-406, as follows:

“It seems likely that both the 2019 and 2020 seasons benefitted from an increased awareness of the YWTMP incentive scheme among hunters since it was first advertised in September 2018 and that time-to-engagement may have been delayed. However, it is not possible to fully disentangle the effect on engagement based on reward versus concerns for cervid health. Increased knowledge of winter ticks and their potential harm to hosts, gained from YWTMP social media and printed materials, may have served to improve participation in the scheme over time. Recognizing the value of community contributions, particularly in cases where participation requires considerable effort in the collection, transportation, and submission of physical samples, not only serves to partly compensate volunteers for their time and efforts but emphasizes the importance of the data collected as part of a larger research program and improved winter tick surveillance. These findings suggest that appealing to the extrinsic motivations of voluntary participants may be particularly worthwhile to boost sample receipt over time, but we suggest this may be most effective in conjunction with improved outreach that equally appeals to intrinsic concerns. In our case, alternative options to immediate physical rewards, such as prize draw entries, could potentially optimize financial costs in future while building capacity for engagement through improved outreach programs in remote areas [54,55].”

4) Lines 433-439: Is there really a legitimate concern about human health and winter ticks and is this a reasonable/honest incentive to promote participation versus 3) above?  This seems a stretch given the remoteness of the area, the probability of human exposure, and that exposure is strictly limited to the questing period of 8-10 weeks in autumn only. 

Although we agree that disease transmission to humans is not a primary concern, it is still one that we believe is worth mentioning here. Research into the potential role of D. albipictus in disease transmission has indicated that the most likely diseases, anaplasmosis and babesiosis, could be transferred via infected larvae (transovarial transmission arising from female ticks becoming infected on host).

However, our reference to disease is not intended to be with respect to participation in hide sampling. We wished to convey the value of tick awareness in general in the north, including other species that would present direct concerns to human health and are occasionally found in the territory (e.g. D. andersoni, known to vector rickettsia and tularemia pathogens).

We have clarified this in the paragraph (now lines 442-447) as follows:

“Wildlife health remains of primary concern with respect to this species, but a growing body of evidence suggests that, in some cases, winter ticks may also be able to vector diseases communicable to humans [63,64]. Additionally, increasing public awareness of ticks of all species in northern communities, particularly among groups at high risk of contact such as including hunters, may serve to improve both tick reporting outside of the YWTMP and commitment to tick bite prevention practices in future [6,8,65].”